# The Mean-Squared Error of Double Q-Learning

**Wentao Weng**
Tsinghua University
wwt17@mails.tsinghua.edu.cn

**Harsh Gupta**
University of Illinois at Urbana-Champaign
hgupta10@illinois.edu

**Niao He**
University of Illinois at Urbana-Champaign
niaohe@illinois.edu

**Lei Ying**
University of Michigan, Ann Arbor
leiying@umich.edu

**R. Srikant**
University of Illinois at Urbana-Champaign
rsrikant@illinois.edu

## Abstract

In this paper, we establish a theoretical comparison between the asymptotic mean-squared error of Double Q-learning and Q-learning. Our result builds upon an analysis for linear stochastic approximation based on Lyapunov equations and applies to both tabular setting and with linear function approximation, provided that the optimal policy is unique and the algorithms converge. We show that the asymptotic mean-squared error of Double Q-learning is exactly equal to that of Q-learning if Double Q-learning uses twice the learning rate of Q-learning and outputs the average of its two estimators. We also present some practical implications of this theoretical observation using simulations.

## 1 Introduction

Reinforcement learning (RL) seeks to design efficient algorithms to find optimal policies for Markov Decision Processes (MDPs) without any knowledge of the underlying model (known as model-free learning) [31]. In this paper, we study the performance of double Q-learning [20, 33], which is a popular variant of the standard Watkins's model-free Q-learning algorithm [34, 35]. Double Q-learning was proposed to remedy the stability issues associated with the standard Q-learning algorithm (due to maximization bias of the Q-function) by using two estimators instead of one. It has been shown empirically that double Q-learning finds a better policy in the tabular setting [20] and converges faster when coupled with deep neural networks for function approximation [33]. Several variations of Double Q-learning were proposed in [36, 1]. However, to the best of our knowledge, there has been no analysis of double Q-learning vis-à-vis how it performs theoretically as compared to standard Q-learning. The objective of this paper is to address this question by providing a tight theoretical comparison between double Q-learning and Q-learning while also drawing experimental insights that allow us to corroborate the theory.

Stochastic Approximation (SA) has proven to be a powerful framework to analyze reinforcement learning algorithms [7, 4, 22]. Several different types of guarantees for various reinforcement learning algorithms have been established using techniques from stochastic approximation. The most commonplace result is the asymptotic convergence of algorithms by analyzing the stability of an associated ODE. Examples include [32], [30] for classical TD-learning with linear function approximation, [8] for synchronous Q-learning, [24] for double TD-learning, and [27, 25] for Q-learning with linear function approximation. To the best of our knowledge, establishing the

convergence of double Q-learning with linear function approximation remains an open problem [24]. Although establishing asymptotic convergence of an algorithm is a useful theoretical goal, quantifying the finite-time convergence rate of an algorithm can be more useful in providing actionable insight to practitioners. There has been a significant body of recent work in this context. Finite-time analyses of TD-learning with either decaying or constant learning rate can be found in [29, 19, 15, 14, 23, 6]. Finite-time error bounds for synchronous Q-learning can be found in [13, 12] and for asynchronous Q-learning in [28]. This line of work primarily focuses on providing upper bounds on the error, thereby failing to make a tight comparison between a pair of algorithms designed for solving the same problem. Recently, several papers developed tight error bounds for SA and RL algorithms, including [16, 17, 11, 21].

In this paper, we focus on comparing Double Q-learning with standard Q-learning, both theoretically and experimentally. We observe that through a particular linearization technique introduced in [16], both Double Q-learning and Q-learning can be formulated as instances of Linear Stochastic Approximation (LSA). We further utilize a recent result [11] that characterizes the asymptotic variance of an LSA recursion by a Lyapunov equation. By analyzing these associated Lyapunov equations for both Q-learning and Double Q-learning, we establish bounds comparing these two algorithms.

The main contributions of this work are two-fold:

**(1) Theoretical Contributions:** We consider asynchronous Double Q-learning and Q-learning with linear function approximation with decaying step-size rules (as special cases of the more general LSA paradigm). Under the assumptions that the optimal policy is unique, both the algorithms converge and the step-size for Double Q-learning is twice that of Q-learning, we show that the asymptotic mean-squared errors of the two estimators of Double Q-learning are strictly worse than that of the estimator in Q-learning, while the asymptotic mean-squared error of the average of the Double Q-learning estimators is indeed equal to that of the Q-learning estimator. This result brings interesting practical insight, leading to our second set of contributions.

**(2) Experimental Insights:** Combining results from our experiments and previous work, we have the following observations:

1. If Double Q-learning and Q-learning use the same step-size rule, Q-learning has a faster rate of convergence initially but suffers from a higher mean-squared error. This phenomenon is observed both in our simulations and in earlier work on variants of Double TD-learning [24].
2. If the step-size used for Double Q-learning is twice that of Q-learning, then Double Q-learning achieves faster initial convergence rate, at the cost of a possibly worse mean-squared error than Q-learning. However, if the final output is the average of the two estimators in Double Q-learning, then its asymptotic mean-squared error is the same as that of Q-learning.

The thumb rule that these observations suggest is that one should use a higher learning rate for Double Q-learning while using the average of its two estimators as the output.

## 2 Q-learning and Double Q-learning

Consider a Markov Decision Process (MDP) specified by $(\mathcal{S}, \mathcal{A}, P, R, \gamma)$. Here $\mathcal{S}$ is the finite state space, $\mathcal{A}$ is the finite action space, $P \in \mathbb{R}^{|\mathcal{S}||\mathcal{A}| \times |\mathcal{S}|}$ is the action-dependent transition matrix, $R \in \mathbb{R}^{|\mathcal{S}| \times |\mathcal{A}|}$ is the reward matrix, and $\gamma \in [0, 1)$ is the discount factor. Upon selecting an action $a$ at state $s$, the agent will transit to the next state $s'$ with probability $P((s, a), s')$ and receive an immediate reward $R(s, a)$.

A policy is a mapping from a state to an action, which specifies the action to be taken at each state. It is well known that the optimal policy can be obtained by solving the so-called Bellman equation [5, 31] for the state-action value function, also called the Q-function:

$$Q^*(s, a) = R(s, a) + \gamma \sum_{s' \in \mathcal{S}} P((s, a), s') \max_{a' \in \mathcal{A}} Q^*(s', a'). \tag{1}$$

In reinforcement learning, the goal is to estimate the Q-function from samples, without knowing the parameters of the underlying MDP. For simplicity, we assume the MDP is operated under a fixed behavioral policy, and we observe a sample trajectory of the induced Markov chain

$\{(S_1, A_1), \cdots, (S_n, A_n), \cdots\}$. Let $X_n = (S_n, A_n)$ and define $\mathcal{X} = \mathcal{S} \times \mathcal{A}$. Since the state space could be fairly large, function approximation is typically used to approximate the $Q$-function. In this work, we focus on linear function approximation for its tractability. The goal is to find an optimal estimator $\theta^* \in \mathbb{R}^d$, such that $Q^* \approx \Phi^\top \theta^*$, where $\Phi = (\phi(s^1, a^1), \cdots, \phi(s^{|\mathcal{X}|}, a^{|\mathcal{X}|})) \in \mathbb{R}^{d \times |\mathcal{X}|}$, and $\phi(s, a) \in \mathbb{R}^d$ are given feature vectors associated with pairs of states and actions.

## 2.1 Q-learning

We first consider asynchronous Q-learning [34, 35] with linear function approximation. Let $\Phi = (\phi(x_1), \cdots, \phi(x_{|\mathcal{X}|})) \in \mathbb{R}^{d \times |\mathcal{X}|}$ be the matrix consisting of columns of feature vectors. We let $\pi_\theta$ denote the greedy policy with respect to the parameter vector $\theta$, i.e., $\pi_\theta(s) = \arg\max_a \phi(s, a)^T \theta$, where we assume that we break ties in the maximization according to some known rule. For ease of notation, we define $H(\theta_1, \theta_2, s) := \phi(s, \pi_{\theta_1}(s))^\top \theta_2$. This function estimates the Q-function based on $\theta_2$ while the action is selected from the greedy policy given by $\theta_1$. When observations on the sample path proceed to $(X_n, S_{n+1})$, Q-learning updates the parameter $\theta$ according to the equation:

$$\theta_{n+1} = \theta_n + \alpha_n \phi(X_n) \left( R(X_n) + \gamma H(\theta_n, \theta_n, S_{n+1}) - \phi(X_n)^\top \theta_n \right), \tag{2}$$

where $\alpha_n$ is an appropriately chosen step-size, also known as the learning rate.

## 2.2 Double Q-learning

To improve the performance of Q-learning, Double Q-learning was introduced in [20, 33]. We consider the Double Q-learning with linear function approximation here. Double Q-learning maintains two estimators $\theta_n^A, \theta_n^B$, which are updated to estimate $Q^*$ based on the sample path $\{X_n\}$ in the following manner:

$$\begin{aligned} \theta_{n+1}^A &= \theta_n^A + \beta_n \delta_n \left( \phi(X_n) \left( R(X_n) + \gamma H(\theta_n^A, \theta_n^B, S_{n+1}) - \phi(X_n)^\top \theta_n^A \right) \right), \\ \theta_{n+1}^B &= \theta_n^B + (1 - \beta_n) \delta_n \left( \phi(X_n) \left( R(X_n) + \gamma H(\theta_n^B, \theta_n^A, S_{n+1}) - \phi(X_n)^\top \theta_n^B \right) \right), \end{aligned} \tag{3}$$

where $\beta_n$ are IID Bernoulli random variables equal to one w.p. $1/2$ and $\delta_n$ is the step-size. Note that at each time instant, only one of $\theta_A$ or $\theta_B$ is updated.

## 2.3 Linear Stochastic Approximation

Under the assumptions that the optimal policy is unique, the ordinary differential equation (ODE) associated with Q-learning is stable and other technical assumptions, it has been argued in [16] that the asymptotic variance of $Q$-learning can be studied by considering the recursion

$$\theta_{n+1} = \theta_n + \alpha_n \phi(X_n) \left( R(X_n) + \gamma \phi(S_{n+1}, \pi^*(S_{n+1}))^\top \theta_n - \phi(X_n)^\top \theta_n \right), \tag{4}$$

where $\pi^*$ is the optimal policy $\pi_{\theta^*}$ based on $\theta^*$. Here and throughout, as in [16], we assume that the Q-learning and Double Q-learning algorithms converge to some $\theta^*$. We refer the reader to [16] for details.

Using a similar argument, one can show that the asymptotic variance of Double Q-learning can be studied by considering the following recursion:

$$\begin{aligned} \theta_{n+1}^A &= \theta_n^A + \beta_n \delta_n \left( \phi(X_n) \left( R(X_n) + \gamma \phi(S_{n+1}, \pi^*(S_{n+1}))^\top \theta_n^B - \phi(X_n)^\top \theta_n^A \right) \right), \\ \theta_{n+1}^B &= \theta_n^B + (1 - \beta_n) \delta_n \left( \phi(X_n) \left( R(X_n) + \gamma \phi(S_{n+1}, \pi^*(S_{n+1}))^\top \theta_n^A - \phi(X_n)^\top \theta_n^B \right) \right). \end{aligned} \tag{5}$$

Our comparison of the asymptotic mean-squared errors of Q-learning and Double Q-learning will use (4)-(5). In practice, however, one is typically interested in how quickly one learns the optimal policy which cannot be measured very well using the mean-squared error metric. Later, we will see that our simulations indicate that the insights we obtain from mean-squared error analysis hold even for learning the optimal policy.

# 3 Main Results

In this section, we present our main results. Before we do, we first review the results on asymptotic variance of linear stochastic approximation in [11] and use these to compare the asymptotic variances of Q-learning and Double Q-learning.

### 3.1 Preliminaries

Consider the linear stochastic approximation recursion:

$$\xi_{n+1} = \xi_n + \frac{g}{n}\left(A(Y_n)\xi_n + b(Y_n)\right), \tag{6}$$

where $g$ is a positive constant, $Y_n$ is an irreducible, aperiodic Markov Chain on a finite state space, $A$ and $b$ are a random matrix and a random vector, respectively, which are determined by $Y_n$. Without loss of generality, we assume $\xi_n$ converges to $\xi^* = 0$. If $\xi^* \neq 0$, we can subtract $\xi^*$ from $\xi_n$. Define the asymptotic covariance of $\xi_n$ to be

$$\Sigma_\infty = \lim_{n\to\infty} n\mathbb{E}\left[\xi_n \xi_n^T\right].$$

The following result is from [11].

**Theorem 1.** *Suppose that $\bar{A} := \mathbb{E}\left[A(Y_\infty)\right]$, and $\frac{1}{2}I + g\bar{A}$ is a Hurwitz matrix, i.e., its eigenvalues have negative real parts, and $\Sigma_b := \sum_{n=2}^{\infty} \mathbb{E}\left[b(Y_n)b(Y_1)^\top\right]$, where $Y_\infty$ is notation for a random variable with the same distribution as the stationary distribution of the Markov chain $\{Y_n\}$. Then, $\Sigma_\infty$ is the unique solution to the Lyapunov equation*

$$\Sigma_\infty\left(\frac{1}{2}I + g\bar{A}^\top\right) + \left(\frac{1}{2}I + g\bar{A}\right)\Sigma_\infty + g^2\Sigma_b = 0. \tag{7}$$

In the next subsection, we use the above result to establish the relationship between the asymptotic covariances of Q-learning and Double Q-learning.

### 3.2 Comparison of Q-learning and Double Q-learning

Throughout this section, we assume that $\theta^* = 0$ without loss of generality. If $\theta^* \neq 0$, the results can hold by subtracting $\theta^*$ from the estimators of Q-learning and Double Q-learning. Our main result is stated in the following theorem.

**Theorem 2.** *Define the asymptotic mean-squared error of Q-learning to be*

$$\mathrm{AMSE}(\theta) := \lim_{n\to\infty} n\mathbb{E}\left[\theta_n^T \theta_n\right],$$

*the asymptotic mean-squared error of the estimator in Double Q-learning to be*

$$\mathrm{AMSE}(\theta^A) := \lim_{n\to\infty} n\mathbb{E}\left[(\theta_n^A)^\top \theta_n^A\right],$$

*and the asymptotic mean-squared error of the average of the two Double Q-learning estimators to be*

$$\mathrm{AMSE}\left(\frac{\theta^A + \theta^B}{2}\right) = \lim_{n\to\infty} \frac{1}{4}n\mathbb{E}\left[(\theta_n^A + \theta_n^B)^\top(\theta_n^A + \theta_n^B)\right].$$

*Let the step sizes of Q-learning and Double Q-learning be $\alpha_n = g/n$ and $\delta_n = 2g/n$, where $g$ is a positive constant. Then there exists some $g_0 > 0$, such that for any $g > g_0$, the following results hold:*

1. $\mathrm{AMSE}(\theta^A) \geq \mathrm{AMSE}(\theta)$, and

2. $\mathrm{AMSE}(\frac{\theta^A + \theta^B}{2}) = \mathrm{AMSE}(\theta)$.

Before we present the proof of the above result, we make some remarks.

**Remark 1.** The condition $g > g_0$ is tied to the sufficient conditions for stability of the ODEs associated with covariance equations of Q-learning and Double Q-learning [11]. If we consider both in tabular case, namely, $\Phi$ is exactly an identity matrix with dimension $|\mathcal{X}|$. Let $\mu_{\min}$ be the minimum probability of a state $x \in \mathcal{X}$ in the stationary distribution $\mu$. In this case, the results hold so long as $g > \frac{1}{\mu_{\min}(1-\gamma)}$, which is a common assumption used in the analysis of tabular Q-learning [28].

**Remark 2.** As mentioned in the introduction to this paper, Double Q-learning can be slower initially due to the fact that only half the samples are used to estimate each of its estimators. One way to speed up the initial convergence rate is to double the learning rate. Our results here show that the

asymptotic mean-squared error of Double Q-learning in that case will be at least as large as that of Q-learning; however, if the output of Double Q-learning is the average of its two estimators, the asymptotic mean-squared error is exactly equal to that of Q-learning with half the learning rate. Thus, Double Q-learning learns faster without sacrificing asymptotic mean-squared error. This suggests that increasing the learning rate of Double Q-learning while averaging the output can have significant benefits, which we verify using simulations in the next section. Now, we are ready to present the proof of the theorem.

**Proof of Theorem 2:** Recall from Section 2.3 that the asymptotic variance of Q-learning can be studied by considering the following recursion:

$$\theta_{n+1} = \theta_n + \alpha_n \phi(X_n) \left( R(X_n) + \gamma \phi(S_{n+1}, \pi^*(S_{n+1}))^\top \theta_n - \phi(X_n)^\top \theta_n \right). \tag{8}$$

Similarly, one can show that the asymptotic variance of double Q-learning can be studied by considering the following recursion:

$$\begin{aligned}
\theta_{n+1}^A &= \theta_n^A + \beta_n \delta_n \left( \phi(X_n) \left( R(X_n) + \gamma \phi(S_{n+1}, \pi^*(S_{n+1}))^\top \theta_n^B - \phi(X_n)^\top \theta_n^A \right) \right), \\
\theta_{n+1}^B &= \theta_n^B + (1 - \beta_n) \delta_n \left( \phi(X_n) \left( R(X_n) + \gamma \phi(S_{n+1}, \pi^*(S_{n+1}))^\top \theta_n^A - \phi(X_n)^\top \theta_n^B \right) \right).
\end{aligned} \tag{9}$$

For ease of notation, let $Z_n = (X_n, S_{n+1})$. It is shown in [13] that $\{Z_n\}$ is also an aperiodic and irreducible Markov chain. Let us define the following: $b(Z_n) = \phi(X_n)R(X_n)$, $A_1(Z_n) = \phi(X_n)\phi(X_n)^\top$, $A_2(Z_n) = \gamma \phi(X_n)\phi(S_{n+1}, \pi^*(S_{n+1}))^\top$, $A(Z_n) = A_2(Z_n) - A_1(Z_n)$. Using these definitions, we can rewrite (8) and (9) as:

$$\theta_{n+1} = \theta_n + \alpha_n \left( b(Z_n) + A_2(Z_n)\theta_n - A_1(Z_n)\theta_n \right). \tag{10}$$

and

$$\begin{aligned}
\theta_{n+1}^A &= \theta_n^A + \beta_n \delta_n \left( b(Z_n) + A_2(Z_n)\theta_n^B - A_1(Z_n)\theta_n^A \right), \\
\theta_{n+1}^B &= \theta_n^B + (1 - \beta_n)\delta_n \left( b(Z_n) + A_2(Z_n)\theta_n^A - A_1(Z_n)\theta_n^B \right),
\end{aligned} \tag{11}$$

respectively. Let $U_n = ((\theta_n^A)^\top, (\theta_n^B)^\top)^\top$. We can further write (11) in a more compact form as:

$$U_{n+1} = U_n + \alpha_n \left[ \begin{pmatrix} -2\beta_n A_1(Z_n) & 2\beta_n A_2(Z_n) \\ 2(1 - \beta_n)A_2(Z_n) & -2(1 - \beta_n)A_1(Z_n) \end{pmatrix} U_n + \begin{pmatrix} 2\beta_n b(Z_n) \\ 2(1 - \beta_n)b(Z_n) \end{pmatrix} \right]. \tag{12}$$

Let $\mu$ denote the steady-state probability vector for the Markov chain $\{X_n\}$. Let $D$ be a diagonal matrix of dimension $|\mathcal{X}|$ such that $D_{ii} = \mu_i$. We have $\bar{A}_1 = \mathbb{E}[A_1(Z_\infty)] = \Phi D \Phi^\top$, $\bar{A}_2 = \mathbb{E}[A_2(Z_\infty)] = \gamma \Phi D P S_{\pi^*} \Phi^\top$, where $S_{\pi^*}$ is the action selection matrix of the optimal policy $\pi^*$ such that $S_{\pi^*}(s, (s, \pi^*(s))) = 1$ for $s \in \mathcal{S}$. Denote $\bar{A} = \bar{A}_2 - \bar{A}_1$.

We will now use Theorem 1 to prove our result. Let $\Sigma_\infty^Q = \lim_{n\to\infty} n\mathbb{E}[\theta_n \theta_n^T]$ and $\Sigma_\infty^D = \lim_{n\to\infty} n\mathbb{E}[U_n U_n^T]$. Clearly, $\text{AMSE}(\theta) = \text{Tr}(\Sigma_\infty^Q)$. Applying Theorem 1 to (10) and (12):

$$\Sigma_\infty^Q \left( \frac{1}{2}I + g\bar{A}^\top \right) + \left( \frac{1}{2}I + g\bar{A} \right) \Sigma_\infty^Q + g^2(B_1 + B_2) = 0, \tag{13}$$

and

$$\Sigma_\infty^D \left( \frac{1}{2}I + g\bar{A}_D^\top \right) + \left( \frac{1}{2}I + g\bar{A}_D \right) \Sigma_\infty^D + g^2 \Sigma_b^D = 0, \tag{14}$$

where $B_1 = \mathbb{E}[\sum_{n=1}^\infty b(X_n)b(X_1)^\top]$, $B_2 = \mathbb{E}[\sum_{n=2}^\infty b(X_n)b(X_1)^\top]$, $\bar{A}_D = \begin{pmatrix} -\bar{A}_1 & \bar{A}_2 \\ \bar{A}_2 & -\bar{A}_1 \end{pmatrix}$, and $\Sigma_b^D = 2\begin{pmatrix} B_1 & B_2 \\ B_2 & B_1 \end{pmatrix}$. Because of the symmetry in the two estimators comprising Double Q-learning, we observe that $\Sigma_\infty^D$ will have the following structure: $\Sigma_\infty^D = \begin{pmatrix} V & C \\ C & V \end{pmatrix}$, where

$$V = \lim_{n\to\infty} n\mathbb{E}[\theta_n^A (\theta_n^A)^\top] = \lim_{n\to\infty} n\mathbb{E}[\theta_n^B (\theta_n^B)^\top], \qquad C = \lim_{n\to\infty} n\mathbb{E}[\theta_n^A (\theta_n^B)^\top].$$

Coupling this observation with (14) yields

$$\begin{pmatrix} V & C \\ C & V \end{pmatrix} + g \begin{pmatrix} V & C \\ C & V \end{pmatrix} \begin{pmatrix} -\bar{A}_1 & \bar{A}_2 \\ \bar{A}_2 & -\bar{A}_1 \end{pmatrix}^T + g \begin{pmatrix} -\bar{A}_1 & \bar{A}_2 \\ \bar{A}_2 & -\bar{A}_1 \end{pmatrix} \begin{pmatrix} V & C \\ C & V \end{pmatrix} + 2g^2 \begin{pmatrix} B_1 & B_2 \\ B_2 & B_1 \end{pmatrix} = 0. \tag{15}$$

Summing the first two blocks (row-wise) of matrices in the above equation, we get

$$V + C + g(V + C)(\bar{A}_2 - \bar{A}_1)^T + g(\bar{A}_2 - \bar{A}_1)(V + C) + 2g^2(B_1 + B_2) = 0. \tag{16}$$

Next, define $g_0 := \inf\{g \geq 0 : g \max(\lambda_{\max}(\bar{A}), \lambda_{\max}(\bar{A}_D)) < -1\}$, where $\lambda_{\max}(A)$ denotes the real part of the maximum eigenvalue of $A$. Note that $g_0$ exists since both $\bar{A}$ and $\bar{A}_D$ are Hurwitz, under the assumption that Q-learning and Double Q-learning both converge [10]. As a result, for any $g > g_0$, $\frac{1}{2}I + g\bar{A}$ is Hurwitz. Therefore, the solution $V + C$ to the above equation and the solution $\Sigma_\infty$ to (13) are unique [10]. Similarly, we also note that the solution to (15) is also unique as $\frac{1}{2}I + g\bar{A}_D$ is Hurwitz whenever $g > g_0$.

Comparing the above equation with (13), we get $\Sigma_\infty^Q = \frac{V+C}{2}$. Next, we observe that $\text{Tr}(V) \geq \text{Tr}(C)$. The reasoning behind that is as follows:

$$\lim_{n\to\infty} n\mathbb{E}\left[(\theta_n^A - \theta_n^B)^T(\theta_n^A - \theta_n^B)\right] \geq 0$$

$$\Rightarrow 2 \lim_{n\to\infty} n\{\mathbb{E}\left[(\theta_n^A)^T\theta_n^A\right] - \mathbb{E}\left[(\theta_n^B)^T\theta_n^A\right]\} \geq 0 \Rightarrow \text{Tr}(V) - \text{Tr}(C) \geq 0,$$

where the second inequality follows from the symmetry in the two estimators comprising double Q-learning. Using $\text{Tr}(V) \geq \text{Tr}(C)$, we get $\text{Tr}(V) \geq \text{Tr}\left(\frac{V+C}{2}\right) = \text{Tr}(\Sigma_\infty^Q)$. This equation proves our first result. To prove the second result, we observe that

$$\text{AMSE}\left(\frac{\theta^A + \theta^B}{2}\right) = \frac{1}{2}\text{AMSE}(\theta^A) + \frac{1}{2}\text{Tr}(C) = \frac{1}{2}(\text{Tr}(V) + \text{Tr}(C)) = \text{Tr}(\Sigma_\infty^Q).$$

$$\square$$

# 4 Numerical Results

In this section, we provide numerical comparisons between Double Q-learning and Q-learning on Baird's Example [2], GridWorld [18], CartPole [3] and an example of maximization bias from [31] [1]. We investigate four algorithms: 1) Q-learning using step size $\alpha_n$, denoted as Q in plots; 2) Double Q-learning using step size $\alpha_n$, denoted as D-Q; 3) Double Q-learning using step size equal to $2\alpha_n$, denoted as D-Q with twice the step size; 4) Double Q-learning using step size equal to $2\alpha_n$ and returning the average estimator $(\theta_n^A + \theta_n^B)/2$, denoted as D-Q average with twice step size. For the vanilla Double Q-learning, we always use $\theta_n^A$ as its estimator.

For the first two experiments, we plot the logarithm of the mean-squared error for each algorithm. We set the step size $\alpha_n = \frac{1000}{n+10000}$. The optimal estimator, $\theta^*$, is calculated by solving the projected Bellman equation [25] based on the Markov chain. Sample paths start in state 1 in Baird's Example, and state $(1, 1)$ in GridWorld. We use the uniformly random policy as the behavioral policy, i.e., each valid action is taken with equal probability in any given state. Initialization of $\theta_1, \theta_1^A, \theta_1^B$ are set the same and are uniformly sampled from $[0, 2]^d$, where $d$ is the dimension of features. Results in each plot reflect the average over 100 sample paths.

## 4.1 Baird's Example

The first environment we consider is the popular Baird's Example which was used to prove that Q-learning with linear function approximation may diverge [13, 2]. It is a simple Markov chain as shown in Fig. 1a with 6 states and 2 actions (represented by the dotted line and the solid line respectively). When the action represented by the dotted line is taken, the agent transits to one of the first five states randomly. When an action represented by a solid line is taken, the agent transits to state 6. The $Q$-function is approximated by a parameter $\theta \in \mathbb{R}^{12}$, where the specific linear combination is shown next to the corresponding action. For the reward function $R(s, a)$, $1 \leq s \leq 6$, $1 \leq a \leq 2$,

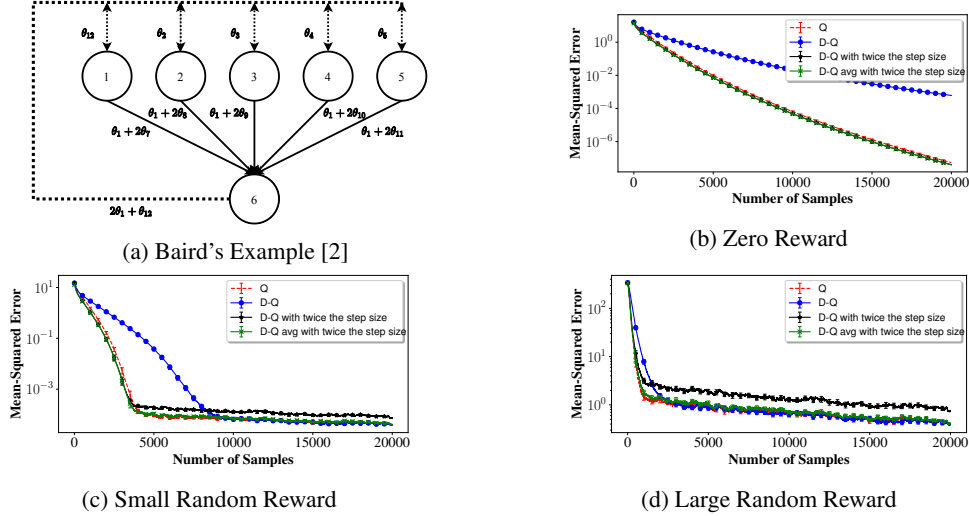

(a) Baird's Example [2]

(b) Zero Reward

(c) Small Random Reward

(d) Large Random Reward

Figure 1: Simulation results for Baird's example. The y-axis is in log scale.

we explore different settings: 1) **Zero Reward:** the reward $R(s,a)$ is uniformly zero; 2) **Small Random Reward:** the reward $R(s,a)$ is sampled uniformly from $[-0.05, 0.05]$; 3) **Large Random Reward:** the reward $R(s,a)$ is sampled uniformly from $[-50, 50]$. Our theory applies to Small Random Reward and Large Random Reward because the optimal policy is unique in these two cases, but simulations indicate that our insight works more generally even in the case of Zero Reward. Although Baird's example was originally proposed to make Q-learning diverge when $\gamma$ is large, we study the case $\gamma = 0.8$ where all algorithms converge. Results are presented in Fig. 1b, 1c, and 1d.

In all the three scenarios, we observe that Double Q-learning converges much slower than Q-learning at an early stage, when using the same step-size . When using a step size $2\alpha_n$, we observe that Double Q-learning converges slightly faster than Q-learning in Fig. 1b, Fig. 1c, and almost at the same speed in Fig. 1d. However, the mean-squared error is much worse than that of Q-learning as shown in Fig. 1c and Fig. 1d. Finally, by simply using the averaged estimator, Double Q-learning obtains both faster convergence rate and smaller mean-squared error, which matches with our theory.

## 4.2 GridWorld

The second environment we simulate is the GridWorld game with a similar setting as in [18]. Consider a $n \times n$ grid where the agent starts at position $(1, 1)$ and the goal is to reach the position $(n, n)$. A $3 \times 3$ GridWorld is shown in Fig. 2a. For each step, the agent can walk in four directions: up, down, left or right. If the agent walks out of the grid, the agent will stay at the same cell. There is a $30\%$ probability that the chosen direction is substituted by any one of the four directions randomly. The agent receives reward $-10^{-3}$ in each step, but receives reward 1 at the destination. The game ends when the agent arrives at the destination. We consider GridWorld with $n = 3, 4$ and 5, so the number of pairs of states and actions can be up to 100. The discount factor is set as $\gamma = 0.9$. We run tabular Q-learning and tabular Double Q-learning. Simulation results are shown in Fig. 2.

As we can see, Double Q-learning using step size $\alpha_n$ converges much slower than all the other three algorithms even though it has a slightly better asymptotic variance as shown in Fig. 2b. By simply doubling the step-size and using the averaged output, Double Q-learning outperforms Q-learning in all the three settings. It is worth pointing out that theoretically speaking, Theorem 2 does not apply to this example because the optimal policy is not unique. However, the insights offered by Theorem 2 still hold.

## 4.3 CartPole

The third experiment we conduct is the classical CartPole control problem introduced in [3]. In this problem, a cart with a pole is controlled by applying a force, either to left or to right. The goal is to keep the pole upright for as long as possible. The player receives a $+1$ reward for every time step

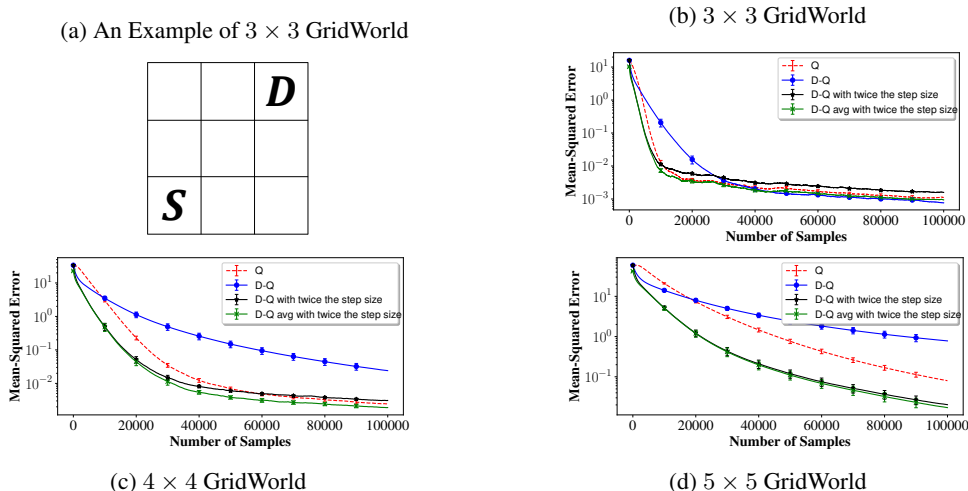

(a) An Example of $3 \times 3$ GridWorld

(b) $3 \times 3$ GridWorld

(c) $4 \times 4$ GridWorld

(d) $5 \times 5$ GridWorld

Figure 2: Simulation results for GridWorld with dimensions $3, 4, 5$. In all the three simulations, Double Q-learning with twice the step-size and averaged output outperforms Q-learning.

until the episode ends which happens when the pole falls down or the cart moves out of a certain region. Unlike the previous numerical results which mainly focus on the mean-squared error, in this case, we study how fast the four algorithms can find a policy that achieves the best performance. We train algorithms on CartPole-v0 available in OpenAI Gym [9]. Specifically, we consider Q-learning and Double Q-learning equipped with $\epsilon$-greedy exploration. The training is episodic, in the sense that for each episode, i.e., the step-size and the $\epsilon$ are updated after one episode. In particular, for the $n$th episode, we use $\epsilon_n = \max(0.1, \min(1, 1 - \log(\frac{n}{200}))), \alpha_n = \frac{40}{n+100}$. The step size is different from previous experiments because we only train 1000 episodes for CartPole, and therefore, the step-size would have remained too large throughout if we had used the previous step-size rule and we noticed that this leads to convergence issues. The discount factor is set as $\gamma = 0.999$. Since the state space of CartPole is continuous, we discrete it into 72 states following [26].

We evaluate the algorithms based on their "hit time", i.e., the time at which they first learn a fairly good policy. We say an algorithm learns a fairly good policy if the mean reward of the greedy policy based on the estimator learned from the first $n$ episodes exceed 195. To reduce the computational overhead, we evaluate the policy obtained after every 50 episodes by averaging the reward obtained by the policy over 1000 independently run episodes. The distribution of the "hit time" for each algorithm in 100 independent tests is shown in Fig. 3. We observe that Double Q-learning using the same learning rate performs much worse than other algorithms. However, when using twice the step size, Double Q-learning finds a good policy faster than Q-learning, at the cost of a larger standard deviation for the "hit time". The increase of variance can be mitigated by using the averaged estimator, which at the same time improves the convergence speed.

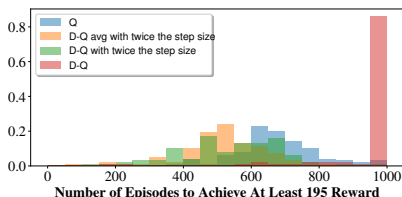

| Algorithm | Mean Hit Time |
|---|---|
| Q | $645.0 \pm 12.93$ |
| D-Q avg with twice the step size | $487.5 \pm 12.19$ |
| D-Q with twice the step size | $518.0 \pm 14.77$ |

Figure 3 & Table 1: Distribution of "hit time", i.e., number of episodes needed to obtain a mean reward of 195 in CartPole-v0, with the number of episodes capped at 1000. The mean hit time of each algorithm is summarized with its standard deviation.

## 4.4 Maximization Bias of Q-learning

The fourth example we investigate is the maximization bias example similar to that in [31, Page 135]. Since Double Q-learning was proposed to alleviate the maximization bias from Q-learning. we study how the proposed modification, doubling the step size and averaging the two estimators in Double

Q-learning, affects the performance in an example where Double Q-learning is known to be helpful. To be specific, there are $M + 1$ states labelled as $\{0, \cdots, M\}$ with two actions, left and right. The agent starts at state $0$. If the agent goes to the right, the game ends, but if she moves to the left, she goes with equal probability to one of the other $M$ states. Both actions result in zero reward. When the agent is at state $1$ to state $M$, if she goes to the right, she returns to state $0$; if she goes to the left, the game ends. Both actions result in a reward independently sampled from a normal distribution with mean $-0.1$ and standard deviation $1$.

We first test the algorithms in a tabular setting with $M = 8$. The exploration policy is set to be $\epsilon$-greedy with $\epsilon = 0.1$. In the $n$th episode, $\alpha_n = \frac{10}{n+100}$. We train the algorithms for $200$ episodes. All estimators are initialized as zero. To evaluate the algorithms, we plot the probability of the agent going left after every episode. In particular, at the end of $n$ episodes, we count how often the estimated Q-function of a left action is larger than that of a right action at state $0$. In addition, the probability is taken to be the average of $1000$ independent runs. Notice that going right always maximizes the mean reward for the agent, so a larger probability to go left indicates that the algorithm has learned a worse policy. The result is shown in Fig. 4a. As we can see, Q-learning suffers from the maximization bias when the number of episodes is small since there is a large probability of going to the left. On the other hand, there is no such problem with Double Q-learning. Furthermore, Double Q-learning with twice the step size and averaging improves performance even more.

In addition to the tabular setting, we also explore a setting where neural networks are used for function approximations. In particular, we consider the same environment as before, but with $M = 10^9$. In this way, it is infeasible to use a table for the whole $Q-$function. We assume that the $Q-$function is approximated by a neural network with two hidden layers of dimension $4$ and $8$. Each pair of adjacent layers is fully connected, with ReLU as the activation function. We use stochastic gradient descent with no momentum as the optimizer. Other settings are the same as those in the tabular setting. The result is shown in Fig. 4b. We can see that although Q-learning does not seem to suffer from maximization bias any more, it performs worse than Double Q-Learning. In addition, Double Q-Learning with twice the step size and averaging helps improve the performance.

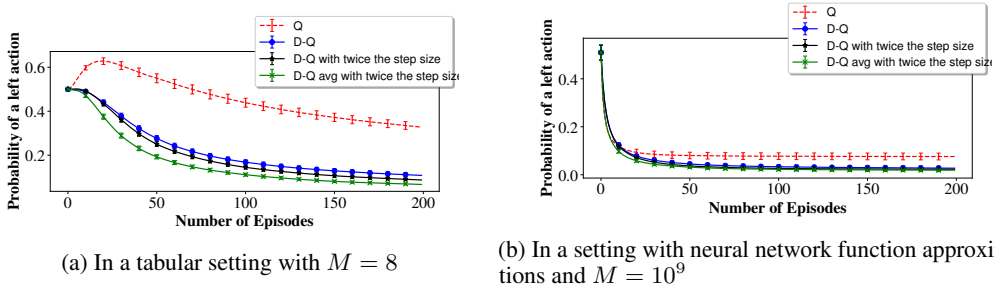

(a) In a tabular setting with $M = 8$

(b) In a setting with neural network function approximations and $M = 10^9$

Figure 4: The probability to go to the left for different algorithms in an environment similar to the maximization bias example from [31]. A lower probability indicates a better policy.

## 5   Conclusion

It is known from prior work that Q-learning has faster convergence rate while Double Q-learning has better mean-squared error. A natural attempt to improve the convergence rate of Double Q-learning is to increase its stepsize (also called learning rate), but this leads to worse mean-squared error. We theoretically showed that increasing the learning rate of Double Q-learning while using a simple averaging at the output improves its convergence rate while making the mean-squared error equal to that of Q-learning. In the supplementary material, we further expand on our theoretical results. Our theoretical results are further supported by numerical experiments which also provide some useful guidelines for implementations of Double Q-learning. However, these results do not immediately apply to Double Q-learning with nonlinear function approximation, which we leave for future investigation.

## Broader Impact

Reinforcement learning (RL) has been the driving force behind many recent breakthroughs in Artificial Intelligence, including defeating humans in games (e.g., chess, Go, StarCraft), self-driving cars, smart home automation, among many others. However, much of the successes build on efficient heuristics and empirical explorations, lacking sufficient theoretical understanding. One such example is Double Q-learning, which is the common practice used in deep reinforcement learning. This work establishes a theoretical analysis of the mean-squared error of double Q-learning, and provides principled guidelines for its implementation. These contributions have the potential to promote a stronger understanding of common RL algorithms both in theory and practice, accelerate the design of more efficient, interpretable RL algorithms, and benefit tremendous RL-driven applications that are societally impactful.

**Acknowledgment:** The work of Wentao Weng was conducted during a visit to the Coordinated Science Lab, UIUC during 2020. Research is also supported in part by ONR Grant N00014-19-1-2566, NSF/USDA Grant AG 2018-67007-28379, ARO Grant W911NF-19-1-0379, NSF Grant CCF 1934986.

## Footnotes

[1]Codes are at `https://github.com/wentaoweng/The-Mean-Squared-Error-of-Double-Q-Learning`.

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
