[Supplementary Material]

# A  Linearization Results

In this section, we provide more details on the derivation of the results pertaining to the asymptotic mean-squared errors in Theorem 2. While [16] provides an outline of the result, we provide some missing details here, including additional assumptions under which the result in [16] is valid. The following result from [4] will be useful to us.

## A.1  Central Limit Theorem for SA

Statements in this part are adapted from [4, Chapter 2 and 3]. Consider a SA algorithm of the form

$$\xi_n = \xi_{n-1} + \gamma_n W(\xi_{n-1}, Y_n), \tag{17}$$

where $\xi_n$ lies in $\mathbb{R}^d$, and the state $Y_n$ lies in $\mathbb{R}^k$. Suppose the algorithm satisfies following assumptions.

**Assumption 1.** *[4, Page 43, Assumption A]*

*(a). Decreasing Step Size:*

$$\gamma_n \geq 0; \ \sum_n \gamma_n = +\infty; \ \sum_n \gamma_n^\alpha < \infty \ \text{ for some } \alpha > 1. \tag{18}$$

*(b). Markovian Noise: There exists a Markov chain $\{\eta_n\}$, independent of $\{\xi_n\}$ with a unique stationary distribution such that $Y_n = f(\eta_n)$.*

*(c). Existence of a Mean Vector Field: We assume the existence of the mean vector field defined by*

$$w(\xi) := \lim_{n\to\infty} \mathbb{E}\left[W(\xi, Y_n)\right],$$

*where the expectation is taken under the distribution of $(Y_n)$.*

Assumption 1(c) allows us to introduce the ODE

$$\dot{\xi} = w(\xi), \xi(0) = z \tag{19}$$

whose unique solution is denoted as $[\xi(z,t)]_{t\geq 0}$. The next assumption we have is on the ODE.

**Assumption 2.** *[4, Assumption (A.2), Assumption (A.2b)] The ODE (19) has an attractor $\xi^*$, whose domain of attraction is denoted by $D_*$. Assumption 1 is satisfied in $D_*$.*

Further, we assume the uniqueness of the attractor.

**Assumption 3.** *[4, Page 108] The ODE is globally asymptotically stable with a unique stable equilibrium point $\xi^*$.*

Define

$$C(\xi) := \sum_{n=-\infty}^{+\infty} \mathrm{Cov}[W(\xi, Y_n), W(\xi, Y_1)] \tag{20}$$

where $\mathrm{Cov}$ denotes the covariance when $Y_1$ is stationary. We can now state the central limit theorem.

**Theorem 3.** *[4, Page 110, Theorem 3] Suppose Assumption 2 and Assumption 3 hold, and the step size sequence satisfies $\gamma_n = \frac{1}{n}$. If $\nabla_\xi w(\xi^*)$ and $C(\xi^*)$ exist, and $\lambda_{\max}(\nabla_\xi w(\xi^*)) < -\frac{1}{2}$, we have*

$$n^{\frac{1}{2}}(\xi_n - \xi^*) \xrightarrow{d} \mathcal{N}(0, P) \tag{21}$$

*where $P$ is the unique symmetric solution of the Lyapunov equation*

$$\left(\frac{I}{2} + \nabla_\xi w(\xi^*)\right) P + P \left(\frac{I}{2} + \nabla_\xi w(\xi^*)^\top\right) + C(\xi^*) = 0.$$

## A.2  Applications to Q-learning and Double Q-learning

In this section, we show that Theorem 3 is applicable to Q-learning (2) and Double Q-learning (3) under the assumptions stated in the main body of the paper. Note that the step sizes are assumed to be $\alpha_n = \frac{g}{n}$, and $\delta_n = \frac{2g}{n}$ in Theorem 2, which are different from that in Theorem 3. Therefore, we

scale the reward function and feature vectors to absorb the constant $g$ (or $2g$) in updates of Q-learning and Double Q-learning. The step sizes are then shifted to $\frac{1}{n}$.

Recall $Z_n = (X_n, S_{n+1})$ defined in the proof of Theorem 2. We first notice that Assumption 1 is automatically satisfied because: 1) The step size condition is fulfilled for $\frac{1}{n}$; 2) The samples $\{Z_n, n \geq 0\}$ form a Markov chain independent of $\theta_n$; 3) The mean vector field $w(\theta)$ is well-defined since $\{Z_n\}$ has a unique limiting stationary distribution, and its state space $\mathcal{X} \times \mathcal{S}$ is finite. As a result, the ODE for Q-learning is defined as

$$\dot{\theta}(t) = g\mathbb{E}\left[\phi(X_n)(R(X_n) + \gamma H(\theta(t), \theta(t), S_{n+1}) - \phi(X_n)^\top \theta(t))\right], \tag{22}$$

and that of Double Q-learning is given by

$$\dot{\theta}^A(t) = g\mathbb{E}\left[\phi(X_n)(R(X_n) + \gamma H(\theta^A(t), \theta^B(t), S_{n+1}) - \phi(X_n)^\top \theta^A(t))\right], \tag{23a}$$

$$\dot{\theta}^B(t) = g\mathbb{E}\left[\phi(X_n)(R(X_n) + \gamma H(\theta^B(t), \theta^A(t), S_{n+1}) - \phi(X_n)^\top \theta^B(t))\right]. \tag{23b}$$

For ease of notation, denote $U(t) = ((\theta^A(t)); (\theta^B(t)))$. The notation $(\mathbf{a}; \mathbf{b})$ is a vector that is the concatenation of $\mathbf{a}$ and $\mathbf{b}$. Also, denote the right hand side of (22) by $w(\theta(t))$, and that of (23) by $\tilde{w}(U(t))$.

To guarantee Assumption 2 and Assumption 3, we make the following assumption.

**Assumption 4.** *Both $\theta(t)$ and $U(t)$ have unique globally asymptotically stable (GAS) equilibrium points.*

Sufficient conditions under which Q-learning with linear function approximation satisfies Assumption 4 are studied in [25, 27]. While little is known on the convergence of Double Q-learning with linear function approximation, it is commonly perceived that double Q-learning is more stable than Q-learning even when equipped with neural networks [33].

Denote the unique stable point of $\theta(t)$ as $\theta^*$, and that of $U(t)$ as $U^*$. It is shown in [27] that $\theta^*$ is the solution to the projected Bellman equation. The following lemma shows that $(\theta^*; \theta^*)$ is also the GAS equilibrium point of the ODE of Double Q-learning. The reader is referred to the next section for the proof.

**Lemma 1.** *The point $U^*$ is exactly $(\theta^*; \theta^*)$.*

To apply Theorem 3, we need to work out $\nabla_\theta w(\theta^*), C_\theta(\theta^*), \nabla_U \tilde{w}(U^*), C_U(U^*)$ which are the analogs of the quantities in (20) for Q-learning and Double Q-learning, respectively. However, since the function $H$ in (22) could be non-differentiable around $\theta^*$, we impose the following assumption from [16] that ensures the existence of $\nabla_\theta w(\theta^*)$ and $\nabla_U \tilde{w}(U^*)$.

**Assumption 5.** *The optimal policy $\pi^* := \pi_{\theta^*}$ is unique.*

Under this assumption, we summarize the exact forms of $\nabla_\theta w(\theta^*), C_\theta(\theta^*), \nabla_U \tilde{w}(U^*), C_U(U^*)$ in the following result. The proof of this lemma is deferred to the next section.

**Lemma 2.** *Following the notation in the proof of Theorem 2, the following equalities hold:*

$$\nabla_\theta w(\theta^*) = g\bar{A}, \quad C_\theta(\theta^*) = g^2(B_1 + B_2); \tag{24a}$$

$$\nabla_U \tilde{w}(U^*) = g\bar{A}_D, \quad C_U(U^*) = 2g^2 \begin{pmatrix} B_1 & B_2 \\ B_2 & B1 \end{pmatrix}, \tag{24b}$$

*where $B_1 := \mathbb{E}\left[\sum_{n=1}^\infty W(Z_n)W(Z_1))^\top\right]$, $B_2 := \mathbb{E}\left[\sum_{n=2}^\infty W(Z_n)W(Z_1)^\top\right]$, and $W(Z_n) := (b(Z_n) + A_2(Z_n)\theta^* - A_1(Z_n)\theta^*)$.*

Note that in Theorem 2, we assume $\theta^* = 0$. Therefore, $W(Z_n) = b(Z_n)$.

Define $g_0 := \inf\{g \geq 0 : g \max(\lambda_{\max}(\bar{A}), \lambda_{\max}(\bar{A}_D)) < -1\}$. Then whenever $g > g_0$, we have $\lambda_{\max}(\nabla_\theta w(\theta^*)) < -\frac{1}{2}, \lambda_{\max}(\nabla_U \tilde{w}(U^*)) < -\frac{1}{2}$. So far we have checked all conditions in Theorem 3 for Q-learning and Double Q-learning. Therefore, the central limit theorem holds:

$$n^{\frac{1}{2}}(\theta_n - \theta^*) \xrightarrow{d} \mathcal{N}(0, P_Q) \tag{25a}$$

$$n^{\frac{1}{2}}(U_n - U^*) \xrightarrow{d} \mathcal{N}(0, P_D) \tag{25b}$$

where $P_Q, P_D$ are given by

$$\left(\frac{I}{2} + g\bar{A}\right) P_Q + P_Q \left(\frac{I}{2} + g\bar{A}^\top\right) + g^2(B_1 + B_2) = 0 \tag{26a}$$

$$\left(\frac{I}{2} + g\bar{A}_D\right) P_D + P_D \left(\frac{I}{2} + g\bar{A}_D^\top\right) + 2g^2 \begin{pmatrix} B_1 & B_2 \\ B_2 & B_1 \end{pmatrix} = 0. \tag{26b}$$

We can see Eq. (26a) and Eq. (26b) are indeed identical to the two equations, Eq. (13) and Eq. (14), for the asymptotic covariance matrices of Q-learning and Double Q-learning. However, since we only establish convergence in distribution of a sequence of random vectors, it does not immediately imply that the limit of variances of these random vectors converges to the variance of the corresponding normal distribution. To fix this gap, we first observe that the function $\mathbf{x}^\top \mathbf{x}$ is continuous where $\mathbf{x}$ is a vector. By the Continuous Mapping Theorem for random vectors and Eq. (25), it holds

$$n\|\theta_n - \theta^*\|_2^2 \xrightarrow{d} \|\mathbf{X}_Q\|_2^2 \tag{27a}$$

$$n\|(U_n - U^*)\|_2^2 \xrightarrow{d} \|\mathbf{X}_D\|_2^2. \tag{27b}$$

where $\mathbf{X}_Q$ follows the normal distribution $\mathcal{N}(0, P_Q)$, and $\mathbf{X}_D$ follows $\mathcal{N}(0, P_D)$. Here, the convergence in distribution is for random variables. Finally, to establish the convergence of the mean of these random variables, we need uniform integrability, which we assume as follows.

**Assumption 6.** *The three sequences of random variables*

$$\{n\|\theta_n - \theta^*\|_2^2, n \geq 1\}, \{n\|\theta_n^A - \theta^*\|_2^2, n \geq 1\}, \{n\|\theta_n^B - \theta^*\|_2^2, n \geq 1\}$$

*are all uniformly integrable.*

Assumption 6 directly implies the sequence $\{n\|U_n - U^*\|_2^2, n \geq 1\}$ is uniformly integrable. Combining (27) with Assumption 6, we have

$$\lim_{n\to\infty} n\mathbb{E}\left[\|\theta_n - \theta^*\|_2^2\right] = \mathbb{E}\left[\|\mathbf{X}_Q\|_2^2\right] = \text{Tr}(P_Q) \tag{28a}$$

$$\lim_{n\to\infty} n\mathbb{E}\left[\|(U_n - U^*)\|_2^2\right] = \mathbb{E}\left[\|\mathbf{X}_D\|_2^2\right] = \text{Tr}(P_D). \tag{28b}$$

Under all the assumptions stated in this section, the linearizations in Section 2.3 are valid.

### A.3 Proof of Lemmas

In this section, we provide missing proofs of Lemma 1 and Lemma 2.

**Proof of Lemma 1:** By Assumption 4, the ODE of Double Q-learning has a unique GAS equilibrium point. Denote this point as $(\theta_1; \theta_2)$. By the symmetry of the ODE (23), $(\theta_2; \theta_1)$ is also a GAS equilibrium point of the ODE. But such point is unique. We thus have $\theta_1 = \theta_2$. In this case, the ODE (23) degenerates to the ODE (22) of Q-learning. Therefore, we have $\theta_1 = \theta_2 = \theta^*$. $\square$

**Proof of Lemma 2:** We show it for Q-learning. The same strategy can be applied to Double Q-learning.

Recall the ODE of Q-learning defined as (22). We know that $\theta^*$ is the unique GAS equilibrium point of this ODE. Recall that the right hand side of (22) is denoted by $w(\theta(t))$. Then at the point $\theta^*$, the following equality holds:

$$w(\theta^*) = g\left(\mathbb{E}\left[\phi(X_n)R(X_n)\right] + \gamma\mathbb{E}\left[\phi(X_n)H(\theta^*, \theta^*, S_{n+1})\right] - \mathbb{E}\left[\phi(X_n)\phi(X_n)^\top\right]\theta^*\right).$$

Note that the optimal policy $\pi^*$ is unique by assumption. We can rewrite $H(\theta^*, \theta^*, S_{n+1})$ as $\phi(S_{n+1}, \pi^*(S_{n+1}))^\top \theta^*$. Then we can see

$$w(\theta^*) = g\left(\mathbb{E}\left[\phi(X_n)R(X_n)\right] + \gamma\mathbb{E}\left[\phi(X_n)\phi(S_{n+1}, \pi^*(S_{n+1}))^\top \theta^*\right] - \mathbb{E}\left[\phi(X_n)\phi(X_n)^\top\right]\theta^*\right) \tag{29}$$

$$= g\mathbb{E}\left[\phi(X_n)R(X_n)\right] + g(\bar{A}_2 - \bar{A}_1)\theta^*. \tag{30}$$

which is the same as the ODE of the linearization (4) at the point $\theta^*$.

Furthermore, since the optimal policy is unique for $\theta^*$, we can define a constant

$$\omega := \min_{(s,a)\in\mathcal{X}:\, a\neq\pi^*(s)} (\phi(s,\pi^*(s))^\top\theta^* - \phi(s,a)^\top\theta^*) > 0$$

be the minimum gap between value functions of optimal actions and non-optimal actions for all states, estimated by $\theta^*$. Let $\epsilon = \frac{\omega}{3\|\Phi\|_1}$. Consider any $\theta \in \mathbb{R}^d$ satisfying $\|\theta - \theta^*\|_\infty \leq \epsilon$. We claim that the greedy policy $\pi_\theta$ is equal to $\pi^*$. To see that it is true, let us fix a state $s \in \mathcal{S}$. For any $a \in \mathcal{A}$ and $a \neq \pi^*(s)$, it holds

$$\phi(s,\pi^*(s))^\top\theta_a - \phi(s,a)^\top\theta_a \geq \phi(s,\pi^*(s))^\top\theta^* - \phi(s,a)^\top\theta^* - 2\left\|\Phi^\top(\theta-\theta^*)\right\|_\infty \geq \omega - \frac{2\omega}{3} > 0.$$

Therefore, $\pi_\theta = \pi^*$. Consequently, for any $\theta$ such that $\|\theta - \theta^*\|_\infty \leq \epsilon$, it holds $w(\theta) = g\mathbb{E}[\phi(X_n)R(X_n)] + g(\bar{A}_2 - \bar{A}_1)\theta$. Therefore, $\nabla_\theta w(\theta^*) = g\bar{A} = g(\bar{A}_2 - \bar{A}_1)$.

For $C_\theta(\theta^*)$, define

$$W(Z_n) := \phi(X_n)R(X_n) + \gamma\phi(S_{n+1},\pi^*(S_{n+1}))\theta^* - \phi(X_n)\phi(X_n)^\top\theta^*.$$

Then by definition,

$$C_\theta(\theta^*) = \sum_{n=-\infty}^{+\infty} \mathbb{E}\left[(gW(Z_n) - w(\theta^*))(gW(Z_1) - w(\theta^*))^\top\right]$$

$$= g^2 \sum_{n=-\infty}^{+\infty} \mathbb{E}\left[(W(Z_n))(W(Z_1))^\top\right]$$

$$= g^2 \left(\sum_{n=1}^{+\infty} \mathbb{E}\left[(W(Z_n))(W(Z_1))^\top\right] + \sum_{n=2}^{+\infty} \mathbb{E}\left[(W(Z_n))(W(Z_1))^\top\right]\right).$$

$\square$

# B   A Stronger Result for the Mean-Squared Error

In this section, we provide a stronger result for the asymptotic mean-squared error of Double Q-learning. Assume that the vector $b(x)$ defined in the proof of Theorem 2 is not the same for all $x \in \mathcal{X}$. Additionally, assume that $\theta^* = 0$. Following the notation in Theorem 2, we have this result.

**Theorem 4.** *Let the step sizes of Q-learning and Double Q-learning be $\alpha_n = g/n$ and $\delta_n = 2g/n$ respectively, where $g$ is a positive constant. With the same constant $g_0$ in Theorem 2, for any $g > g_0$, it holds*

$$\mathrm{AMSE}(\theta^A) \geq \mathrm{AMSE}(\theta) + c_0 g$$

*where $c_0$ is a positive constant independent from $g$.*

Theorem 4 shows that in general, the asymptotic mean-squared error of Double Q-learning is worse than that of Q-learning, when using twice of the step size. Moreover, the gap scales at least linearly with respect to the step size.

To prove Theorem 4, we need two additional lemmas. The first lemma is on the relationship between the two matrices $\bar{A}_D$ and $\bar{A}$ defined in the proof of Theorem 2.

**Lemma 3.** *Following the notation in the proof of Theorem 2, consider the matrix $\bar{A}_D = \begin{pmatrix} -\bar{A}_1 & \bar{A}_2 \\ \bar{A}_2 & -\bar{A}_1 \end{pmatrix}$. The set of its eigenvalues is given by the union of eigenvalues of $\bar{A}_2 - \bar{A}_1$ and that of $-(\bar{A}_2 + \bar{A}_1)$.*

**Proof of Lemma 3:**   Suppose $\lambda$ is an eigenvalue of $\bar{A}_D$ with an eigenvector $v = (v_1^\top, v_2^\top)^\top \neq 0$ where $v_1, v_2 \in \mathbb{R}^d$. We claim that $\lambda$ is either an eigenvalue of $-\bar{A}_1 + \bar{A}_2$ or an eigenvalue of $-(\bar{A}_1 + \bar{A}_2)$. To see this fact, it holds

$$\bar{A}_D \begin{bmatrix} v_1 \\ v_2 \end{bmatrix} = \lambda \begin{bmatrix} v_1 \\ v_2 \end{bmatrix}.$$

If $v_1 + v_2 \neq \mathbf{0}$, then
$$(-\bar{A}_1 + \bar{A}_2)(v_1 + v_2) = \lambda(v_1 + v_2),$$
showing that $\lambda$ is an eigenvalue of $-\bar{A}_1 + \bar{A}_2$. Otherwise, suppose $v_1 + v_2 = \mathbf{0}$. Then $v_1 = -v_2$, and
$$-(\bar{A}_1 + \bar{A}_2)v_1 = \lambda v_1.$$

We can also show that for every eigenvalue of $-\bar{A}_1 + \bar{A}_2$ and $-(\bar{A}_1 + \bar{A}_2)$, we can construct a corresponding eigenvector with respect to $\bar{A}_D$. Therefore, the set of eigenvalues of $\bar{A}_D$ is exactly the union of eigenvalues of $-\bar{A}_1 + \bar{A}_2$ and $-(\bar{A}_1 + \bar{A}_2)$. $\qquad\square$

The second lemma is on the trace of the solution of a Lyapunov equation.

**Lemma 4.** *Consider a Lyapunov equation*
$$AX + XA^\top + Q = 0,$$

*where $A, Q \in \mathbb{R}^{n \times n}$ are given, for some positive integer $n$. If $A$ is Hurwitz, and $Q \succcurlyeq 0$, and $\mathrm{Tr}(Q) > 0$, then $\mathrm{Tr}(X) > 0$.*

Note that the notation $Q \succcurlyeq 0$ means that $Q$ is a positive semi-definite matrix.

**Proof of Lemma 4:** By [10, Theorem 5.6], if $A$ is Hurwitz, then $X$ has a unique solution that can be expressed as
$$X = \int_0^\infty e^{At} Q e^{A^\top t} \, \mathrm{d}t. \tag{31}$$

Since $Q \succcurlyeq 0$ by assumption, and $(e^{At})^\top = e^{A^\top t}$ for all $t$, we have $X \succcurlyeq 0$. We prove $\mathrm{Tr}(X) > 0$ by contradiction. Suppose $\mathrm{Tr}(X) = 0$. Therefore, as $X \succcurlyeq 0$, we have: $\mathbf{v}^\top X \mathbf{v} = 0, \forall$ vectors $\mathbf{v}$ (since all eigenvalues of $X$ are 0).

Denote the largest eigenvalue of $Q$ as $\lambda_m$, which must be a positive real value because $Q \succcurlyeq 0$ and $\mathrm{Tr}(Q) > 0$. Suppose $\mathbf{v}$ is the unit eigenvector corresponding to $\lambda_m$, i.e., $Q\mathbf{v} = \lambda_m \mathbf{v}$, and $\|\mathbf{v}\|_2 = 1$. We have
$$\mathbf{v}^\top X \mathbf{v} = \int_0^\infty \mathbf{v}^\top e^{At} Q e^{A^\top t} \mathbf{v} \, \mathrm{d}t. \tag{32}$$

Note that $\lim_{t \to 0} e^{At} = I$, and $\lim_{t \to 0} e^{A^\top t} = I$. Therefore, for $\epsilon = \min\left(\frac{\lambda_m}{\|Q\|_2}, 1\right)$, there exists a $\tilde{t} > 0$, such that for any $0 \leq t \leq \tilde{t}$, we have
$$\left\| e^{At} - I \right\|_2 \leq \epsilon, \quad \left\| e^{A^\top t} - I \right\|_2 \leq \epsilon. \tag{33}$$

Equation (32) can be rewritten as
$$\mathbf{v}^\top X \mathbf{v} = \int_0^{\tilde{t}} \mathbf{v}^\top e^{At} Q e^{A^\top t} \mathbf{v} \, \mathrm{d}t + \int_{\tilde{t}}^\infty \mathbf{v}^\top e^{At} Q e^{A^\top t} \mathbf{v} \, \mathrm{d}t \tag{34}$$
$$\overset{a}{\geq} \int_0^{\tilde{t}} \mathbf{v}^\top e^{At} Q e^{A^\top t} \mathbf{v} \, \mathrm{d}t \tag{35}$$
$$= \int_0^{\tilde{t}} \mathbf{v}^\top (I + e^{At} - I) Q (I + e^{A^\top t} - I) \mathbf{v} \, \mathrm{d}t \tag{36}$$
$$= \int_0^{\tilde{t}} \mathbf{v}^\top Q \mathbf{v} \, \mathrm{d}t + \int_0^{\tilde{t}} \mathbf{v}^\top (e^{At} - I) Q \mathbf{v} \, \mathrm{d}t + \int_0^{\tilde{t}} \mathbf{v}^\top Q (e^{A^\top t} - I) \mathbf{v} \, \mathrm{d}t$$
$$+ \int_0^{\tilde{t}} \mathbf{v}^\top (e^{At} - I) Q (e^{A^\top t} - I) \mathbf{v} \, \mathrm{d}t. \tag{37}$$

Inequality $a$ follows from the fact that $e^{At} Q e^{A^\top t} \succcurlyeq 0$, for any $t \geq 0$. To lower bound (37), we first have $\int_0^{\tilde{t}} \mathbf{v}^\top Q \mathbf{v} \, \mathrm{d}t = \tilde{t} \|v\|_2^2 \lambda_m$, by definition of $\mathbf{v}$. For the last three terms, using the definition of

matrix norm and (33), the following hold

$$\left|\int_0^{\tilde{t}} \mathbf{v}^\top (e^{At} - I) Q \mathbf{v} \, dt\right| \leq \tilde{t} \|v\|_2^2 \|Q\|_2 \, \epsilon \tag{38}$$

$$\left|\int_0^{\tilde{t}} \mathbf{v}^\top Q (e^{A^\top t} - I) \mathbf{v} \, dt\right| \leq \tilde{t} \|v\|_2^2 \|Q\|_2 \, \epsilon \tag{39}$$

$$\left|\int_0^{\tilde{t}} \mathbf{v}^\top (e^{At} - I) Q (e^{A^\top t} - I) \mathbf{v} \, dt\right| \leq \tilde{t} \|v\|_2^2 \|Q\|_2 \, \epsilon^2. \tag{40}$$

Therefore, we have

$$
\begin{aligned}
\mathbf{v}^\top X \mathbf{v} &\geq \tilde{t} \|v\|_2^2 \lambda_m - 2\tilde{t} \|v\|_2^2 \|Q\|_2 \, \epsilon - \tilde{t} \|v\|_2^2 \|Q\|_2 \, \epsilon^2 \\
&\geq \tilde{t} \|v\|_2^2 \left(\lambda_m - \|Q\|_2 \left(2\epsilon + \epsilon^2\right)\right) \\
&\geq \frac{1}{2} \tilde{t} \|v\|_2^2 \lambda_m
\end{aligned}
\tag{41}
$$

by the definition of $\epsilon$. We can see that $\mathbf{v}^\top X \mathbf{v} > 0$, which contradicts the assumption that $\mathbf{v}^\top X \mathbf{v} = 0$. Therefore, $\text{Tr}(X) > 0$ by contradiction. $\qquad\square$

We now present the proof of Theorem 4.

**Proof of Theorem 4:** This proof follows the notation in the proof of Theorem 2. In particular, we assume that the random vector $b(X_n)$ is centered at $0$. Recall Eq. (15). Subtracting the block on the upper left corner by that on the upper right corner, we have

$$(V - C) \left(\frac{1}{2}I - g(\bar{A}_1 + \bar{A}_2)\right)^\top + \left(\frac{1}{2}I - g(\bar{A}_1 + \bar{A}_2)\right)(V - C) + 2g^2(B_1 - B_2) = 0. \tag{42}$$

By the definition of $B_1$ and $B_2$, we have $B_1 - B_2 = \mathbb{E}\left[b(X_1)b(X_1)^\top\right]$, whose trace is positive by assumptions. As in the proof of Theorem 2, set the constant $g_0 := \inf\{g \geq 0 : g\max(\lambda_{\max}(\bar{A}), \lambda_{\max}(\bar{A}_D)) < -1\}$. Since the matrix $\bar{A}_D$ is defined as $\begin{pmatrix} -\bar{A}_1 & \bar{A}_2 \\ \bar{A}_2 & -\bar{A}_1 \end{pmatrix}$, we know by Lemma 3, the set of eigenvalues of $-(\bar{A}_1 + \bar{A}_2)$ is a subset of eigenvalues of $\bar{A}_D$. Therefore, for $g > g_0$, we have $g\lambda_{\max}(-(\bar{A}_1 + \bar{A}_2)) < -1$. It immediately implies $\frac{1}{2}I - g(\bar{A}_1 + \bar{A}_2)$ is Hurwitz. Utilizing Lemma 4, we have $\text{Tr}(V - C) > 0$. Together with the result $V + C = 2\Sigma_\infty^Q$ in the proof of Theorem 2, we have

$$\text{AMSE}(\theta^A) = \text{Tr}(V) = \text{Tr}(\Sigma_\infty^Q) + \frac{\text{Tr}(V - C)}{2} > \text{Tr}(\Sigma_\infty^Q) = \text{AMSE}(\theta).$$

On the other hand, to show $\text{AMSE}(\theta^A) - \text{AMSE}(\theta)$ indeed scales up linearly with respect to $g$, we divide both sides of Eq. (42) by $g$

$$(V - C) \left(\frac{1}{2g}I - (\bar{A}_1 + \bar{A}_2)\right)^\top + \left(\frac{1}{2g}I - (\bar{A}_1 + \bar{A}_2)\right)(V - C) + 2g(B_1 - B_2) = 0.$$

Since $\frac{1}{2g}I - (\bar{A}_1 + \bar{A}_2)$ is Hurwitz, the following equation has a unique positive definite solution $X$.

$$X \left(\frac{1}{2g}I - (\bar{A}_1 + \bar{A}_2)\right)^\top + \left(\frac{1}{2g}I - (\bar{A}_1 + \bar{A}_2)\right) X + (B_1 - B_2) = 0$$

Therefore, $\text{Tr}(V - C) = 2g\text{Tr}(X)$. Further, let $X'$ be the solution to the following Lyapunov equation

$$X' \left(-(\bar{A}_1 + \bar{A}_2)\right)^\top + \left(-(\bar{A}_1 + \bar{A}_2)\right) X' + (B_1 - B_2) = 0.$$

Since $-(\bar{A}_1 + \bar{A}_2)$ is Hurwitz, and $B_1 - B_2$ has a positive trace, we have $\text{Tr}(X') > 0$, which is independent of $g$. By the expression Eq. (31) of $X$ and $X'$, it can be easily shown that $\text{Tr}(X) \geq \text{Tr}(X')$. This proves that $\text{AMSE}(\theta^A) - \text{AMSE}(\theta) \geq c_0 g$ for some positive constant $c_0$ independent from $g$. $\qquad\square$