[Reviews · NeurIPS 2020]

Review 1

Summary and Contributions: The authors provide a theoretical analysis of Double Q-learning, specifically the asymptotic MSE in the case of linear function approximation. Their analysis suggests a way to select the step size and final output to obtain faster initial convergence while maintaining the same asymptotic result, which they then verify empirically in different experiments. The asymptotic analysis relies on a stochastic approximation result, which they apply in a regime where the policy to select the next step Q evaluation is already optimal (see Eq 4-5). They can then transform the q-learning and Double Q-learning updates in a similar form (eq 10 and 12) which allows the result to apply and the asymptotic MSE to be compared between the approaches. The main concern here is that by assuming that the max action is fixed to being the optimal policy, we are already in a regime where overestimation bias isn’t present in Q-learning due to the usual reasons (taking expectation after the max), and so Double Q-learning has little to offer and all to lose (since it has split updates and parameters). The asymptotic analysis doesn’t seem like the right way to understand the potential benefits of Double Q-learning in this respect. The authors do point out that by averaging final parameters for Double Q-learning and doubling the step size, Double Q-learning’s AMSE matches Q-learning’ MSE, which then allow the possibility of Double Q-learning helping in the early stages while provably not hurting in the final stages of learning. The potential theoretical benefits of Double Q-learning early on are not elucidated theoretically in this paper, but they do provide some empirical evidence that this story holds. The empirical evidence consists of experiments in 3 domains: Baird’s example, a gridworld, and cartpole. In each domain, experimental results give support to the analysis and the suggestion above about doubling stepsize + averaging. Something I thought was missing in the experiments is again coming back to the issue of estimation bias. These domains may not induce estimation bias for Q-learning (at least this isn’t reported here), and in that case we would not expect Double Q learning to have a large advantage in the early stages of learning. It would have been nice to see a domain with clear estimation bias (e.g. Figure 6.5 in Sutton&Barto’s RL book 2nd edition) and see how that relates to final results for these different methods. Despite my concerns above, the paper is clearly written, the analysis is novel with some interesting empirical result to support it. I will wait for the reviewer's response to decide on final score. -------- post author response --------- Thank you for the additional experiments and clarifications. I have updated my score though I think the paper stands between a 6 and 7 given the limitations of the theoretical analysis. If the paper is accepted, I would appreciate if the authors clarify in the paper these limitations for the benefits of future readers.

Strengths: See above.

Weaknesses: See above.

Correctness: As far as I could tell, the analysis is sound and the results correct.

Clarity: The paper is well-written with clear notation which makes the math easy to follow.

Relation to Prior Work: Relevant past work is well covered as far as I could tell.

Reproducibility: Yes

Additional Feedback:


Review 2

Summary and Contributions: A theoretical comparison between double Q-learning and Q-learning is performed, for the tabular/linear RL setting. In particular, the conditions are studied for which the asymptotic mean-squared error are equal for the two methods. In addition, experiments are performed that study the speed of learning for several settings of these methods. ---response to rebuttal----- I appreciate the additional experiments. However, I'm keeping my current score for two reasons: 1) the improvement shown in the deep RL setting is really very minor, 2) the deep RL results are on a very simple task and do not appear to use an advanced optimizer; I'm skeptical whether any improvement can be shown at all in more complex domains.

Strengths: 1. The authors provide theoretical insights as to why double Q-learning should use a higher step-size than Q-learning, in case of linear/tabular RL. 2. A minor improvement of double Q-learning is proposed (averaging the estimators), for tabular/linear RL.

Weaknesses: The main weakness is that the contribution of the paper is small for a number of reasons: 1. Only tabular/linear RL is considered. However, the main empirical advantage (and usage) of double Q-learning occurs in the deep RL setting. In the deep RL setting, double DQN can result in more stable learning and higher asymptotic performance; with tabular/linear RL convergence is less of an issue and double Q-learning is hardly ever used. Furthermore, it is unclear how easily these results would carry over to deep RL, when modern optimizers such as Adam are used. 2. The main conclusion is that the step-size should be twice as high for double Q-learning compared to regular Q-learning. However, as the step-size is one of the main hyperparameters of a tabular/linear RL method, it is typically tuned per method anyway. So an empirical researcher using double Q-learning will likely quickly realize that it requires a larger step-size than Q-learning. 3. The assumption for the theory that the optimal policy should be unique is quite a strong assumption that limits the impact of this work further.

Correctness: As far as I can tell, the claims appear to be correct.

Clarity: The paper is very well written.

Relation to Prior Work: There are several improvements proposed in the literature of double Q-learning. It would be good to at least acknowledge this and ideally motivate why these are not considered. For example: > "weight double Q-learning", Zhang et al. > "Averaged-DQN: Variance Reduction and Stabilization for Deep Reinforcement Learning" , Anschel et al.

Reproducibility: Yes

Additional Feedback: My main suggestion for improvement of this work is to add deep RL experiments. Even if the theoretical results don't naturally carry over, it would be interesting to see if the suggestions based on the theory for the tabular/linear (larger step-size, averaging estimators) setting carry over to the deep RL setting. This is especially important because double Q-learning is not much used in the tabular/linear setting, for reasons mentioned above. For tabular RL, I would explore & discuss the effects of the Q-value initialization on the advantage of double Q-learning over Q-learning. I suspect that in tasks where optimistic initialization is effective, the bias in the update of Q-learning might actually be an advantage.


Review 3

Summary and Contributions: This paper investigates the convergence speed of Double Q-learning compared to Q-learning as a function of the learning rate. It also proposed an adaptation of double Q-learning which uses the average of the two estimators as the final estimate. The main finding is that double Q-learning obtains the same mean squared error as Q-learning assuming that it uses twice the learning rate and that the two estimators are averaged for the final answer.

Strengths: Double Q-learning is a commonly used algorithm for solving RL problems, so understanding the convergence properties is highly relevant. The paper undertakes extensive theoretical analysis and also makes some algorithmic contributions (averaging the two networks), which get validated with simple experiments.

Weaknesses: The proofs in this paper are limited to tabular policies and linear function approximation, making them less applicable to Deep RL, which is probably a majority of the current applications. It would be great to at least show some results on harder environments to validate the claims. Lastly, I would be interested to understand how the proof would have to be adapted to account for non-unique optimal policies. If this is intractable, an experimental exploration would have been interesting. The experimental section also contains some relatively arbitrary hyperparamters, eg. the exact equation for the step size. It would have been good to confirm whether results are robust for different choices of these parameters. The experimental results also miss the error-of-the-mean, eg. in the cartpole experiments.

Correctness: I did not find any obvious issues with the paper, but also didn't check the proofs in detail.

Clarity: Yes, the paper is overall well written. I would have hoped for a bit more explanation around the proofs, providing the reader with more intuitive understanding of what is happening.

Relation to Prior Work: Yes.

Reproducibility: Yes

Additional Feedback: Update: I have read the author response and am still believe that this paper can provide value to the NeurIPS community.


Review 4

Summary and Contributions: This paper sheds light on special properties of the asymptotic mean square error (AMSE) of Double Q-learning and standard Q-learning, under a linear approximation of the Q-function. Using results from Linear Stochastic Approximation the authors show that the asymptotic covariance of each estimator of the parameter vector satisfies a specific Lyapunov equation. The structures of the Lyapunov equations and asymptotic covariance matrices imply bounds on the AMSE of the different estimators. In particular it is shown that the average of the two estimators given by Double Q-learning with twice the learning rate of standard Q-learning has the same AMSE as the estimator of standard Q-learning. These theoretical results are supported in the experimental section and provide practical guidelines on how to combine the estimators of standard Q-learning and the average of the estimators of Double Q-learning.

Strengths: The paper is well written and easy to read. The exposition of the results the authors borrow is clear and honest. The proofs seem sound, simple and clean. The theoretical findings are indeed supported by well described experiments. Some code is provided in supplementary (I managed to run cartpole.py) The practical guidelines seem useful in practice.

Weaknesses: The methodology presented in the paper applies only when considering a linear approximation of the Q-function. The code provided as supplementary material is written in both Matlab and Python, this is not convenient.

Correctness: The proofs seem sound and not complicated given the exposition of the results the authors borrow.

Clarity: Yes, it is indeed a strength of the paper.

Relation to Prior Work: Relation to prior work and tools borrowed from Linear Stochastic Approximation are clearly exposed in the introduction, Theorem 1 and proof of Theorem 2.

Reproducibility: Yes

Additional Feedback: l 19 vis-à-vis l 27 TD(0) notation is not defined l 88 "This function estimates the value function" -> the Q-function ? l 151 Section 2.3 not clickable l 160 - (12) U_n notation is not defined l 200 spacing Fig.__1a l 368 adA?pted **EDIT** I am satisfied with the rebuttal and discussion, my overall score remains the same.

[Author Response · NeurIPS 2020]

We thank all the reviewers for their comments and constructive feedback.

**Response to Reviewer 1:** *Theoretical results:* The reviewer is correct in that some assumptions are needed to derive
the results. However, we believe that our results are interesting for two reasons: (i) our paper is the first to present
a theoretical analysis of Double Q-learning versus Q-learning that goes beyond asymptotic convergence. (ii) the
experimental results (including additional ones we have conducted) support our theory. In particular, as noted in the
paper, some of our experiments (e.g., Gridworld) are for models which do not satisfy the assumptions needed for the
theory.

*Experimental results:* The suggestion to add experiments for the Sutton-Barto example is very interesting. We have
indeed performed those simulations now, see Figures (a) and (b). To be consistent with Sutton and Barto, for this
example, we have plotted the probability of going left as a function of the number of episodes. Double Q-learning with
twice the step-size and averaging does indeed perform better as the theory suggests. We will include the results for
these experiments along with the implementation details in the final version of the paper.

**Response to Reviewer 2:** *Theoretical results:* Even though the theory does assume a unique optimal policy, our
experimental results indicate that our conclusions hold more broadly. More importantly, we would like to point out that
our paper is the first to show that Q-learning will perform strictly better than Double Q-learning (see Theorem 4 in the
supplementary material which further strengthens the result in Theorem 2) at least under some nontrivial conditions.
Our primary goal is to obtain a theoretical understanding of when and why Double Q-learning performs well. We hope
that our idea of analyzing the asymptotic covariance will stimulate further research leading to a deeper understanding of
Double Q-learning and its variants.

*Experiments beyond linear/tabular settings:* We have now conducted experiments where we have used a neural network
to estimate the Q-function. One such result is shown in Figure (b) for the same example suggested by Reviewer 1
since it has estimation bias. And again Double Q-learning with twice the step size and averaging performs the best,
especially in the initial episodes. We will include the results for these experiments along with the implementation details
in the final version of the paper. The comments about Adam/other step-size selection techniques and initialization are
interesting, we will explore them in the final version (if the paper is accepted) and/or a longer version of the paper.

*References:* We will add and discuss the suggested references in the paper.

**Response to Reviewer 3:**

*Experimental validation of the theory:* As mentioned, Figures (a) and (b) present experimental results on an example
with estimation bias, where we have also used a neural network for Q-function approximation (Figure (b)). The
experimental results in the paper also include examples with non-unique optimal policies. We will include more
experimental results in the paper to show robustness with respect to choice of hyper-parameters and also conduct
additional experiments in complex RL environments. In the cart pole experiment, we have used the number of episodes
to reach a certain reward as our metric and we have the distribution of this quantity. This appears to us to be stronger
than knowing just the errors of the means.

*Clarity:* Thanks for the suggestion to include more explanations, we will do so in the final/longer version of the paper.

**Response to Reviewer 4:**

*Experiments:* Thanks for suggestion about not using code in multiple languages. We will migrate them over to Python
before the conference if the paper is accepted. We have now conducted experiments for cases where a neural network is
used to approximate the Q-function. Studying such a model theoretically is currently a significant challenge in the field.

(a) Sutton-Barto Tabular            (b) Sutton-Barto Neural Network

[Meta-Review · NeurIPS 2020]

There was much discussion regarding the significance of the results and whether these will be relevant to future research. As such, the authors are encouraged to further discuss the technical implications of their result in a revised version, to clarify why it is important, in particular to the deep reinforcement learning setting. Otherwise, there was general consensus that there is something technically novel and sound here.